# Modification of *N*-terminal α-amine of proteins via biomimetic *ortho*-quinone-mediated oxidation

Siyao Wang[1,6], Qingqing Zhou[1,6], Xiaoping Chen[2,6], Rong-Hua Luo[3,4,6], Yunxue Li[1], Xinliang Liu[1], Liu-Meng Yang[3,4], Yong-Tang Zheng [3,4,7✉] & Ping Wang [1,5,7✉]

Naturally abundant quinones are important molecules, which play essential roles in various biological processes due to their reduction potential. In contrast to their universality, the investigation of reactions between quinones and proteins remains sparse. Herein, we report the development of a convenient strategy to protein modification via a biomimetic quinone-mediated oxidation at the *N*-terminus. By exploiting unique reactivity of an *ortho*-quinone reagent, the α-amine of protein *N*-terminus is oxidized to generate aldo or keto handle for orthogonal conjugation. The applications have been demonstrated using a range of proteins, including myoglobin, ubiquitin and small ubiquitin-related modifier 2 (SUMO2). The effect of this method is further highlighted via the preparation of a series of 17 macrophage inflammatory protein 1β (MIP-1β) analogs, followed by preliminary anti-HIV activity and cell viability assays, respectively. This method offers an efficient and complementary approach to existing strategies for *N*-terminal modification of proteins.

[1] Shanghai Key Laboratory for Molecular Engineering of Chiral Drugs, School of Chemistry and Chemical Engineering, Frontiers Science Center for Transformative Molecules, Shanghai Jiao Tong University, Shanghai, China. [2] Key Laboratory of Optoelectronic Devices and Systems of Ministry of Education and Guangdong Province College of Physics and Optoelectronic Engineering, Shenzhen University, Shenzhen, China. [3] Key Laboratory of Animal Models and Human Disease Mechanisms of Chinese Academy of Sciences, Kunming Institute of Zoology, Chinese Academy of Sciences, Kunming, Yunnan, China. [4] Center for Biosafety Mega-Science, Kunming Institute of Zoology, Chinese Academy of Sciences, Kunming, Yunnan, China. [5] Key Laboratory of Systems Biomedicine (Ministry of Education), Shanghai Center for Systems, Shanghai Jiao Tong University, Shanghai, China. [6] These authors contributed equally: Siyao Wang, Qingqing Zhou, Xiaoping Chen, Rong-Hua Luo. [7] These authors jointly supervised this work: Yong-Tang Zheng, Ping Wang. ✉email: zhengyt@mail.kiz.ac.cn; wangp1@sjtu.edu.cn

Oxidation reactions play important roles in organic chemistry and are widely involved in crucial biological transformations[1,2]. In line with the broad interest in the development of selective and mild oxidation reactions, *ortho*-quinone cofactors of copper amine oxidases (CuAOs) have been extensively studied due to their high catalytic efficiency. In the catalytic cycle, the Tyr side chain is converted into lysyl tyrosylquinone, which can readily oxidize primary amines into aldehydes via a quinone-mediated transamination pathway using $O_2$ as a co-oxidant to complete the catalytic cycle (Fig. 1a)[3,4]. Recently, significant progress has been achieved toward the design of quinone-based catalysts with major contributions from Corey[5], Fleury[6], Kobayashi[7], Stahl[8–10] and Luo (Fig. 1b)[11–13]. These biomimetic quinone-based catalysts show specific chemoselectivity toward the dehydrogenation of primary, secondary, tertiary amines or other reactions[14–16]. The applications of quinone oxidation have been largely limited to small molecule transformations despite the significant research progress in this area[17,18]. On the other side, in nature, the reactions between *o*-quinones such as DOPA[19] and protein side chains are important but complex in fundamental process of life[20]. Although the early work provided by Mason in 1955 showed that *o*-quinones could react with proteins via the *N*-terminal residue[21], the reaction between *o*-quinone motifs and proteins have not been fully elucidated on a molecular level[22]. Nevertheless, the intrinsic design of quinones by nature may provide an effective opportunity to modify proteins under physiological conditions.

Existing methods used to modify proteins are mainly confined to nucleophilic amino acids, such as Cys and Lys, and potentially result in heterogeneous conjugates due to the high frequency of these amino acids[23–25]. In this context, selective modification of the *N*-terminus of proteins has led to the single site functionalization of proteins[26–30]. Although significant progress has been made, the majority of these methods required specific *N*-terminal residue, owing to their assistance of the side-chain functionality, such as β-nucleophilically-functionalized Cys[31–35], Ser/Thr[36,37]. Transamination reactions under physiological conditions have been elegantly developed by Francis and co-workers utilizing pyridoxal-5-phosphate (PLP)[38–45] or *N*-methylpyridinium-4-carboxaldehyde benzenesulfonate salt (Rapoport's salt)[46] as

oxidants, which can convert the *N*-termini of proteins into aldo or keto functionalities for oxime formation or other bioconjugation reactions[47–49]. However, the rate or efficiency of transamination/oxime formation may vary at different *N*-termini or even different amino acid combinations[50,51]. Room for improvements is still remained in the field of *N*-terminal modification strategy[52]. Marginally, the salt nature of $NaIO_4$, PLP or RS salt makes them difficult to be removed from aqueous medium after transamination of peptide or small protein targets, which may potentially interfere the subsequent conjugations[53]. Thus, given the generality of quinone and its derivatives in the effective oxidation of amines, we seek to investigate whether or not the quinones may be suitable reagents for the selective and efficient oxidation of the N-terminal α-amines of proteins to aldehyde or ketone with fast kinetics as well as a wide scope of amino acids.

In this work, we show a selective and rapid method for modifying the *N*-termini of proteins via a quinone-mediated oxidation of *N*-terminal α-amine of complex peptides and proteins under physiological conditions (Fig. 1c). Several examples have been demonstrated using a range of peptides and proteins, including ubiquitin, myoglobin, E3 ubiquitin-protein ligase RNF4 (32-133) fragment and small ubiquitin-related modifier 2 (SUMO2). Moreover, we prepared a library of macrophage inflammatory protein 1β (MIP-1β)[54] analogs using a combination of native chemical ligation and quinone-mediated transamination, where the late-stage modification of the *N*-terminus of MIP-1β can lead to a 20-fold increase in its anti-HIV-1 activity.

## Results and discussion

**Development of *ortho*-quinone-mediated oxidation.** To begin our journey, a range of quinone derivatives **Ox1–Ox8** were prepared containing different functionalities with an aim toward fine-tuning their reactivity to be selective for the α-amine rather than the ε-amine of Lys, as well as other functionalities[7,11,12]. We tested the transamination reaction of model peptide **1** (1 mM) with the sequence of GFHAKGY in an aqueous solution buffered at pH 6.5 using 10 mM (saturated) of the quinones (**Ox1–Ox8**) as the oxidants (Fig. 2a). The oxidized product was subsequently reacted with $EtONH_2$ to form its corresponding oxime **1a**. 4-*tert*-

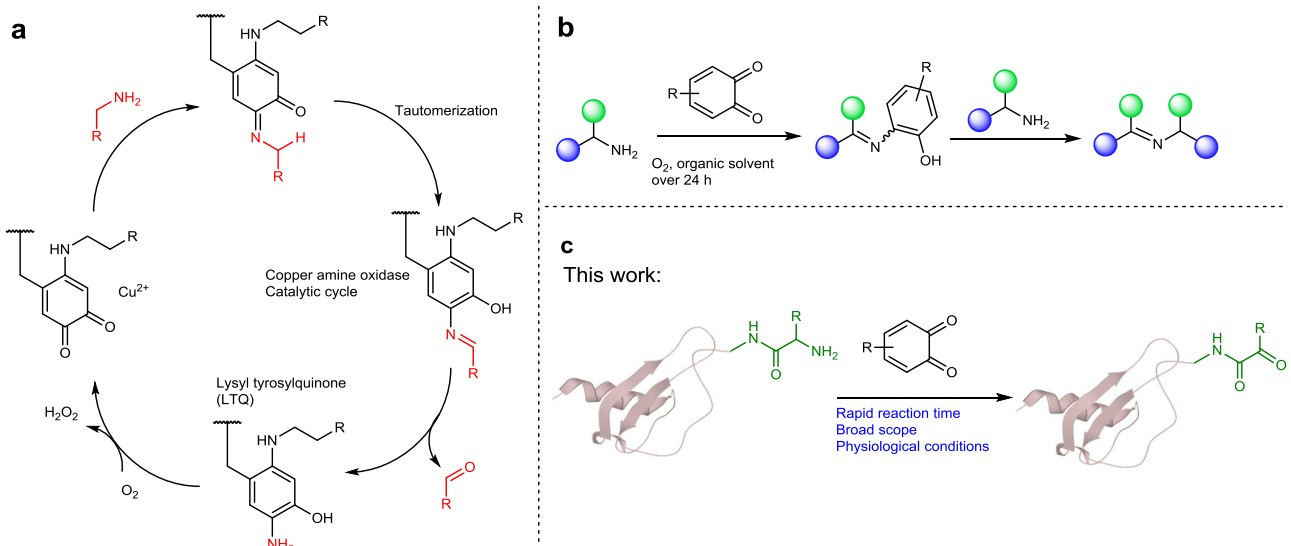

**Fig. 1 Development of site-specific *N*-terminal protein modification via quinone-mediated transamination. a** Natural catalytic cycle of CuAOs; **b** Oxidation of primary amines with *ortho*-quinone to form imines; **c** Biomimetic quinone-mediated transamination inspired by CuAOs. the quinones may be suitable reagents for the selective and efficient oxidation of the *N*-terminal α-amines of proteins to aldehyde or ketone with fast kinetics as well as a wide scope of amino acids.

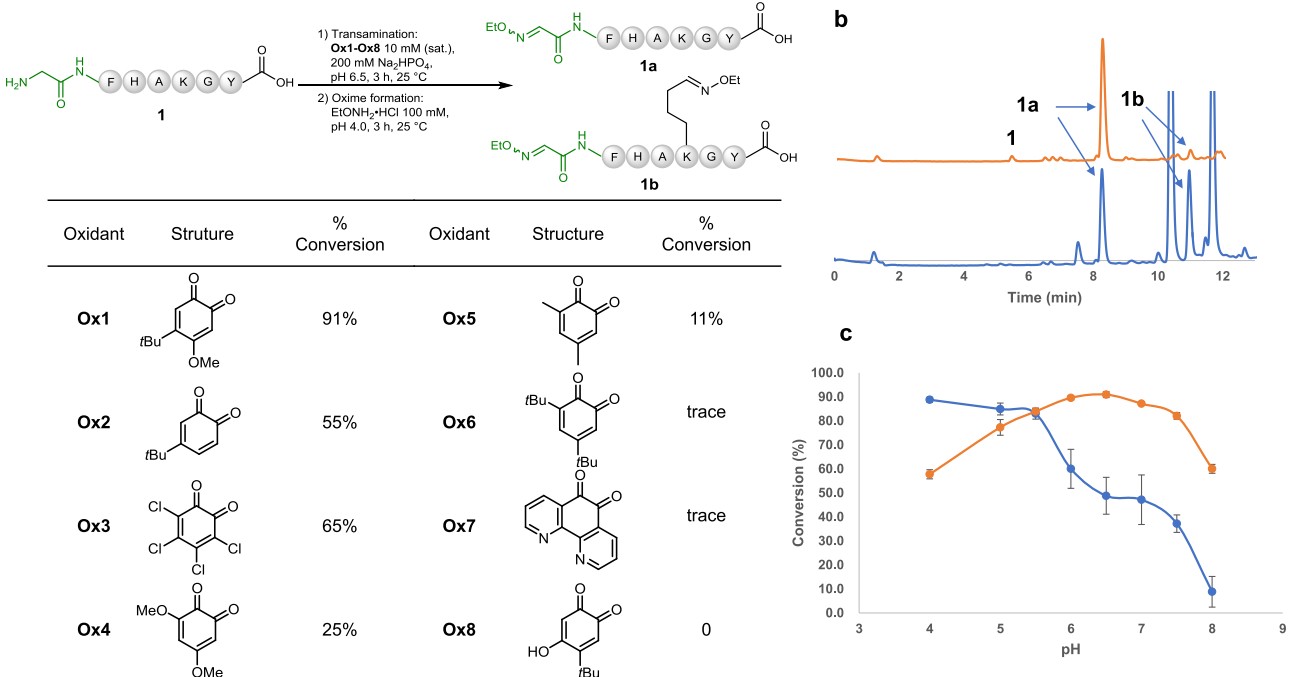

**Fig. 2 Optimization of the reaction conditions used in the biomimetic transamination reaction. a** Screening of oxidants **Ox1-Ox8** used for the transamination reaction with model peptide **1**; **b** LC trace obtained for the transamination reaction using **Ox1** in yellow or **Ox2** in blue; **c** pH Optimization of the transamination reaction using **Ox1** in yellow or **Ox2** in blue (the rate of conversion was calculated by integrating the related peaks of the LC trace measured at 280 nm. Centers represent means, error bars represent standard deviation, $n = 3$ independent replicates).

Butyl-5-methoxy-4-*tert*-butyl-*o*-benzylquinone **Ox1** achieved near quantitative conversion (>90%) to generate product **1a** in 3 h with <2% Lys oxidation observed via LCMS due to the extraordinary low pKa of the *N*-terminal amine compared to that of the Lys side chain amine[26]. 4-*tert*-Butyl-*o*-benzylquinone **Ox2** could also achieve 55% conversion at pH 6.5; the major by-product **1b** was attributed to the undesired Lys side chain oxidation (>40%) (Fig. 2b). Further optimization of the pH of the transamination reactions with either quinone **Ox1** or **Ox2** showed that **Ox1** had an optimal pH range of 6.0–6.5, while the side reaction during using quinone **Ox2** could be minimized by lowering reaction pH to 4.0 (90% conversion to give target product **1a**) (Fig. 2c). Although tetrachloro-*o*-quinone **Ox3** resulted in 65% conversion to desired oxime **1a**, the reaction also led to other side reactions which complexed the system. Other quinone derivatives were also evaluated, but no promising results were given. Thus, **Ox1** serves as the most effective oxidant (see Supplementary Note 8 for the discussion of the mechanism). A time-course monitoring study was performed to verify that the reaction reached to completion in 3 h at room temperature (see Supplementary Note 3.2.3.). Although an early work demonstrated that the $Cu^{2+}$ salt was necessary for *N*-terminal oxidation with glyoxylate as reagent[55], addition of $Cu^{2+}$ in this reaction did not increase the yield of the oxidative product. Interestingly, compared with such reactions with small molecules (Fig. 1b) where an imine was formed between starting amine and ketone, quinone-mediated transamination can readily push towards completion (90% conversion) without formation of such imine derivative which may potentially quench the reaction.

Next, to investigate the scope of the *N*-terminal residue, we expanded the *N*-terminal residue of peptide **1** to the remaining 19 amino acid residues **2**−**19**. As it shows in Fig. **3**, Arg peptide **2**, Leu peptide **3**, Met peptide **4**, Glu peptide **5** and Lys peptide **6** reacted cleanly and gave satisfactory conversion (60–95%) in 3 h to afford the desired products **2a**−**6a** without any significant

observation of the expected side-products. Asp peptide **7** underwent oxidative decarboxylation during the oxidation and therefore formed the methylated product **11a**. Ser peptide **8**, Thr peptide **9** and Trp peptide **10** were oxidized, but all formed the side-chain cleaved oxime product **1a** cleanly. Peptides **11**−**15** (containing Ala, Phe, Ile, Tyr and Val as *N*-termini, respectively) could also be converted into their desired oxime products, but with lower conversions. Asn peptide **16** was oxidized in the first step, but the resulting ketone product could not be converted to oxime effectively in 3 h. Further extension of oxime reaction time to 5 h led to the formation of product **16a** with 88% conversion. Although Gln peptide **17** could be transaminated, it tended to form pyroglutamate (see Supplementary Note 3.3) as a naturally occurred post-translational modification, which significantly affected the efficiency of the overall reactions. His peptide **18** failed to be converted into desired oxime due to the **Ox1** remaining bonded to the peptide. The resulted byproduct (>60% conversion) was confirmed to be an oxazine adducts **18b** (Fig. 3b)[56]. Pro peptide **19** was hardly oxidized as a secondary amine, only trace amount (<2%) of oxime product **19a** was observed. Cys peptide **20** decomposed during reaction. Oxime products **1a**, **4a**, **5a**, **6a**, **11a**, **24a** and byproduct **18b** generated from selected model peptide reactions were carefully characterized with 1D and 2D NMR to confirm the site-selectivity as well as side reactions (see Supplementary Note 2 and 3).

**Transamination of proteins with different *N*-termini.** After screening the scope of *N*-terminal residues in the reaction, we further applied this method to more complex therapeutic peptides and proteins. The first target chosen was COVID-19 spike protein 319-347 fragment, which is located in its receptor binding domain. The peptide contains an Arg residue at its *N*-terminus and a cysteine residue, which is protected with acetamidomethyl (Acm) group. Due to the low solubility of this peptide, the transamination step was carried out in 2 M Gn·HCl buffer for 6 h

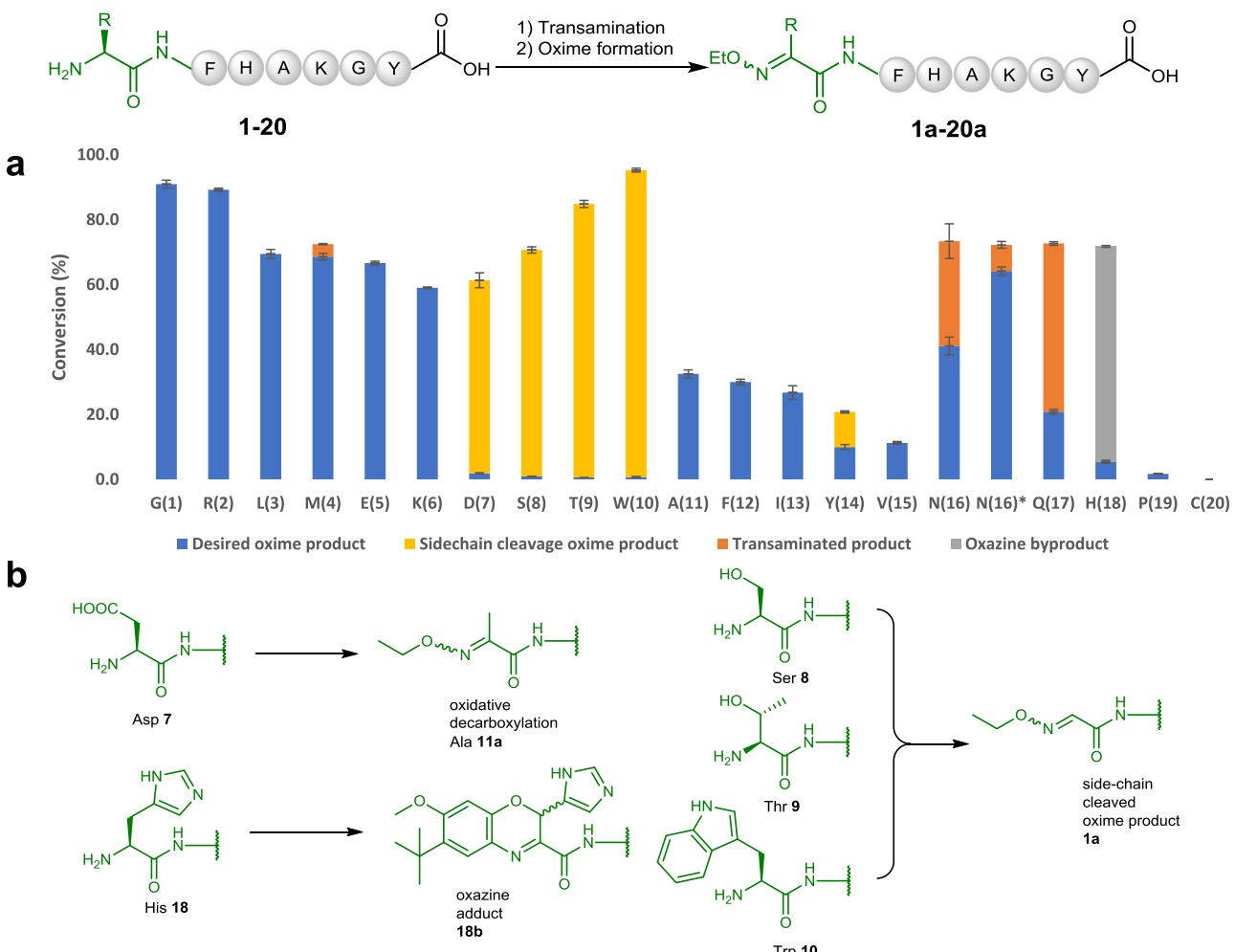

**Fig. 3 Transamination of model peptides with different *N*-termini. a** Landscape for the transamination of 20 canonical amino acids attached on the *N*-terminal residue of the model peptides (Error bars represent standard deviation, $n = 3$ independent replicates). Reaction conditions: (1) Transamination: **Ox1** (10 mM, sat.), 200 mM $Na_2HPO_4$, pH 6.5, 3 h, 25 °C; (2) Oxime formation: $EtONH_2 \cdot HCl$ (100 mM), pH 4.0, 3 h, 25 °C (oxime formation time for **N(16)**\*: 5 h; **b** A summary of major unexpected products of transamination at different *N*-termini with **Ox1**. Structures of these products were characterized with 2D NMR.

with all the other reaction conditions unchanged. The reaction proceeded smoothly to afford the oxime product **21** with 85% conversion (Fig. 4a). Tetracosactide, a therapeutic peptide bearing an *N*-terminal Ser, one Met and four Lys residues in its sequence, was successfully oxidized under the standard reaction conditions in 3 h. The desired oxime product **22** was formed with 80% conversion monitored by LCMS (Fig. 4b). The scope of this transformation was next examined using myoglobin, which contains 18 Lys residues in the sequence (Fig. 4c). This reaction was conducted in aqueous buffer at pH 6.0 to minimize over-oxidation of the Lys side chain and the reaction time was extended to 5 h. Due to the limitation of the chromatographic techniques used, total ion counts (TICs) were used to determine the conversion rate after a simple workup procedure (see Supplementary Note 3.6.2). The desired ethyl oxime product **23a** was observed with 70% conversion, 15% unreacted starting material and 15% of the over-oxidized by-product. Furthermore, a biotinylated tag was synthesized and incorporated to myoglobin. However, due to the low reactivity of this tag, an excess of **Ox1** could not be quenched immediately, which then led to further oxidation during the oxime formation step. To address this issue,

once the first transamination reaction was completed, an ether extraction step was carried out to remove the excess oxidant. The resulting solution underwent the oxime formation step overnight to afford a 40% conversion of biotinylated protein **23b**. In nature, many proteins are expressed with Met as their *N*-terminus. Therefore, we examined this transformation with ubiquitin, which contains an *N*-terminal Met. In 2 M Gn·HCl buffer, ubiquitin was smoothly converted into its oxime product **24** with 77% conversion (Fig. 4d). Although disulfide bonds within peptides and proteins could stay untouched in our reaction, free Cys residue remained challenged as it could form an adduct with **Ox1**. Therefore, we designed a temporary protecting strategy for free Cys containing proteins. We tested the compatibility of this strategy with RNF-4 fragment (32–133, *N*-terminal Glu) and A1G-SUMO2 protein (with mutation of *N*-terminal Ala to Gly) as examples (Fig. 5). Initially, three Cys residues in RNF4 and one in SUMO2 were protected with 2-nitropiperonyl bromide under pH 6.5 for 2 h[57], leading to protected RNF4 **25** and SUMO2 **26**. Next, both proteins underwent transamination with **Ox1**, followed by subsequent oxime formation with $EtONH_2$ (0.1 mM) to form the desired oxime products. After the **Ox1** mediated

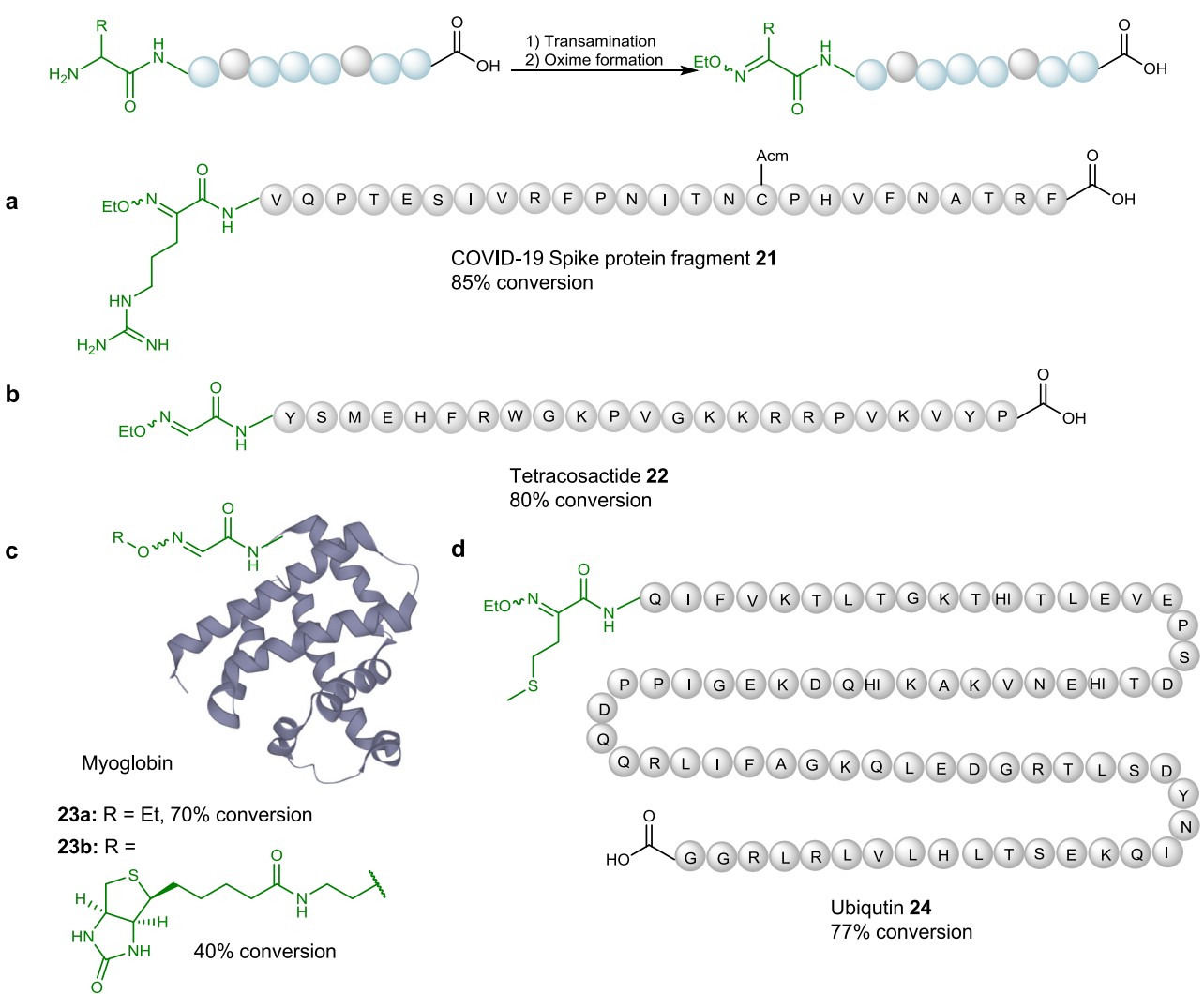

**Fig. 4 Site selective peptide modification via quinone-mediated transamination reaction. a** Reaction conditions for COVID-19 spike protein fragment **21**: (1) Transamination: 2 M Gn·HCl, **Ox1** (10 mM, sat.), 200 mM $Na_2HPO_4$, pH 6.5, 6 h, 25 °C; (2) Oxime formation: $EtONH_2$·HCl (100 mM), pH 4.0, 3 h, 25 °C. **b** Reaction conditions for tetracosactide **22**: (1) Transamination: **Ox1** (10 mM, sat.), 200 mM $Na_2HPO_4$, pH 6.5, 3 h, 25 °C; (2) Oxime formation: $EtONH_2$·HCl (100 mM), pH 4.0, 3 h, 25 °C; **c** Reaction conditions for myoglobin **23a-b**: (1) Transamination: **Ox1** (10 mM, sat.), 200 mM $Na_2HPO_4$, pH 4.0, 5 h, 25 °C; (2) Oxime formation for **23a**: $EtONH_2$·HCl (100 mM), pH 4.0, 3 h, 25 °C; oxime formation for **23b**: extraction with EtOAc (2 ×1.0 mL), then biotinylated tag **S5** (100 mM), pH 6.5, 16 h, 25 °C. **d** Reaction conditions for ubiquitin **24**: (1) Transamination: 2 M Gn·HCl, **Ox1** (10 mM, sat.), 200 mM $Na_2HPO_4$, pH 6.5, 3 h, 25 °C; (2) Oxime formation: $EtONH_2$·HCl (100 mM), pH 4.0, 3 h, 25 °C.

oxidation, the protecting groups were removed tracelessly within 30 min in the presence of 20 nM sodium ascorbate and UV ($\lambda =$ 365 nm) irradiation, affording *N*-terminal modified RNF4 32-133 fragment **27** and SUMO2 **28** with 75% and 80% conversion over three steps, respectively. Circular dichroism spectra of *N*-terminal modified protein **28** was compared with that of the expressed SUMO2 protein to confirm that its tertiary structure remained unchanged during reactions (see Supplementary Note 3.9.2).

**Application of quinone oxidation to medicinal chemistry**. With all the examples above, we confirmed that quinone **Ox1** was able to oxidize peptides or proteins with various *N*-termini. Therefore, it can serve as an alternative strategy to existing methods and be used to expand the application of transamination in biological research. Considering the fast kinetics and wide *N*-terminal tolerance of our transamination reaction, we applied this methodology to a medicinal target, MIP-1β. MIP-1β (CCL4) exhibits potent anti-HIV-1 activity[58] by binding to the hydrophobic

transmembrane helix bundle of CCR5 via its *N*-terminal domain[59,60]. Due to the increasing viral resistance and inefficient therapies, the discovery of alternative HIV-1 inhibitors is still highly coveted[61]. Diversification of the *N*-terminal region of CCLs can be greatly beneficial toward the development of future peptide HIV-1 inhibitors[62–64]. However, current methods rarely facilitate such direct modifications of its *N*-termini[65]. Thus, a merging of late-stage oxidation and state-of-the-art peptide ligation method will allow the fast generation of a protein library for the rapid screening of potential drugs. We synthesized a truncated MIP-1β 4-96 protein **29**, which contained Gly4 and the full-length form of MIP-1β **30** with the mutation of Ala1 to Gly for oxidation (Fig. 6b). Both proteins were chemically synthesized via native chemical ligation[66,67] and refolded using cysteine/cystine redox[68] buffer on a > 30 mg scale (see Supplementary Note 4 for the syntheses of MIP-1β variants **29** and **30**).

Having established a facile protocol to prepare MIP-1β proteins on a large-scale, we next performed the protein *N*-terminal transamination-oxime ligation reaction to modify the

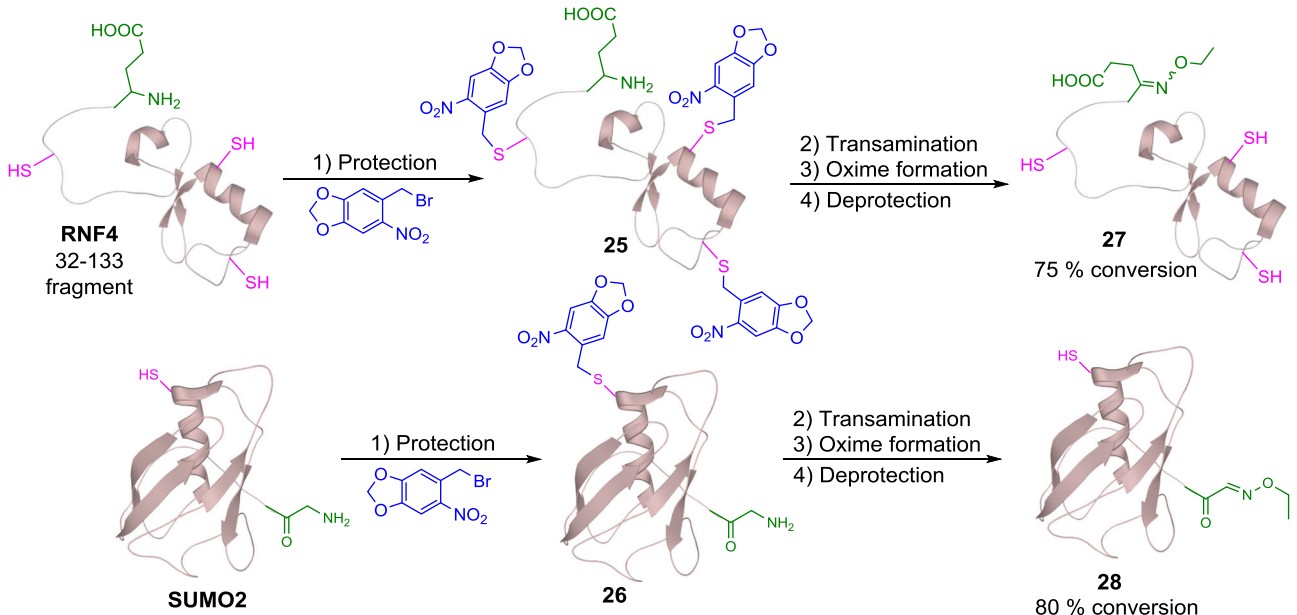

**Fig. 5 Quinone-mediated transamination reaction with proteins bearing free-Cys residues.** Reaction conditions for (1) photolabile protecting group installation: 2-nitropiperonyl bromide (100 equiv., sat.), 200 mM $Na_2HPO_4$, pH 6.5, 2 h, 25 °C; (2) Transamination: **Ox1** (10 mM, sat.), 200 mM $Na_2HPO_4$, pH 6.5, 25 °C, 6 h for RNF4 **25**; 3 h for SUMO2 **26**; (3) Oxime formation: $EtONH_2·HCl$ (100 mM), pH 4.0, 3 h, 25 °C; (4) Deprotection: $hv$ ($\lambda = 365$ nm), $EtONH_2·HCl$ (100 mM, from last step), sodium ascorbate (20 mM), 30 min, 25 °C.

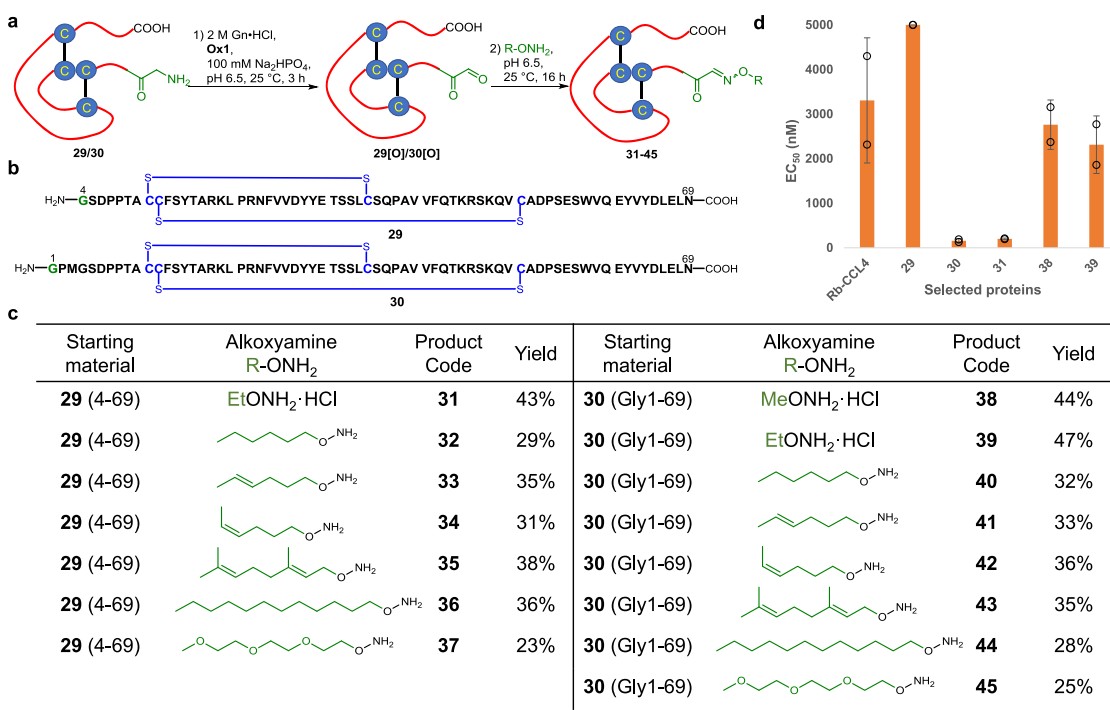

**Fig. 6 Late-stage modification of MIP-1β proteins *via* quinone-mediated selective oxidation of *N*-terminal amine. a** Reaction scheme for the transamination of MIP-1β protein **29** and **30**; **b** The sequences of protein **29** and **30**; **c** A summary of the transamination/oxime ligation of protein **29** and **30** and the isolated yields for all reactions; **d** Anti-HIV-1 activity of selected MIP-1β analogs with $EC_{50} < 5000$ nM (Error bars represent standard deviation, $n = 2$ biological independent replicates).

protein. Various alkyl alkoxyamines were chosen to optimize the anti-HIV-1 activity of MIP-1β according to their hydrophobicity (Fig. 6a). Hexyl, (*E*)-hexyl, (*Z*)-hexyl, geranyl and decyl hydroxylamine were used to investigate the influence of the hydrocarbon chains with different conformation or chain lengths toward fine-tuning the bioactivity (Fig. 6c). In addition, we also synthesized PEGylated hydroxylamine as a hydrophilic tag for

comparison. **Ox1** was added into the target protein **29** or **30** in buffer solution at pH 6.5 and the solution was incubated at 25 °C for 3 h. Subsequently, after the addition of the alkoxyamine, the mixture was readjusted to pH 6.5 and reacted for a further 16 h to afford modified proteins **31**-**45** (Fig. 6c). MIP-1β proteins and $MeONH_2·HCl$ or $EtONH_2·HCl$ were dissolved well in this buffer solution, and the ligations proceeded smoothly and gave high

isolated yields (43% for **31**, 44% for **38** and 47% for **39**, respectively). Reactions with fatty alkoxyamines became biphasic due to their poor solubility, vigorous stirring was necessary during these reactions. After 24 h of reaction, a small amount of starting material was still present (~20% by HPLC analysis), which suggested that these lipo-hydroxylamines did not mix well even in an aqueous buffer and hence resulted in a decreased yield (28–38%). PEGylated variants (**37** and **45**) were obtained with low yielding (23% and 25%, respectively) due to incomplete formation of oxime. Transamination with **Ox1** was fully compatible with the disulfide bonds in these proteins without disrupting their biological activity, as shown by HPLC-MS and the anti-HIV-1 assay.

With all the modified MIP-1β variants in hand, we performed an anti-HIV-1 activity assay to evaluate their inhibitory activity. Recombinant human MIP-1β (Rh-CCL4) was used as the primary standard with a measured half maximal effective concentration ($EC_{50}$) value of 3.3 μM (Fig. 6d). Protein **29** (4-69) resulted in a complete loss of anti-HIV-1 activity within the concentration range studied (up to 5 μM). Interestingly, ethylated variant **31** regained its activity with an $EC_{50}$ value of 200 nM. Full-length Gly1-69 protein **30** also displayed increased inhibitory activity with an $EC_{50}$ value of 157 nM upon substitution of Ala1 with Gly, which suggested that a branched methyl group decreases the binding affinity between MIP-1β and CCR5. Further extension of full-length protein **30** with methoxyamine and ethoxyamine showed comparable activities against HIV-1 by an order of magnitude (**38**: $EC_{50} = 2.7$ μM and **39**: $EC_{50} = 2.3$ μM), whereas all the lipo-variants and PEGylated variants exhibited no activity within the concentration range studied (all $EC_{50} > 5$ μM, see SI). Besides, a cell viability assay with MTT was also performed to explore the cytotoxicity of the synthetic proteins toward TZM-bl cells. In our investigations, all the synthetic proteins exhibited cytotoxicity with 50% cytotoxic concentration ($CC_{50}$) values of > 50 μM (see Supplementary Note 7), which indicated that the proteins showed no cytotoxic behavior. In our future studies, we will further optimize the N-terminal domain with hydroxyamines bearing other functionalities and thereafter explain the exact mechanism of how these modified proteins exhibit their inhibitory activity against HIV-1 both in vitro and in vivo.

In summary, we have demonstrated a highly selective and mild method for modifying the N-terminus of protein via a biomimetic quinone-based transamination reaction. The key features of this reaction include rapid reaction time, high selectivity and a broad scope of amino acids. Covid-19 spike protein fragment (Arg), tetracosactide (Ser), myoglobin (Gly), ubiquitin (Met), RNF4 32-133 fragment (Glu) and SUMO2 (Gly) along with a range of model peptides were given as examples. Furthermore, the preparation of several MIP-1β analogs via native chemical ligation and late-stage N-terminal modification has demonstrated its potential utility in medicinal chemistry. We believe this method will be a useful tool for protein modification and medicinal chemistry studies.

## Methods

**General procedure for transamination with *ortho*-quinone 1.** Peptide (1 μmol) was dissolved in an aqueous solution or 2 M Gn·HCl solution buffered with $Na_2HPO_4$ at pH 6.5, followed by the addition of **ox1** (10 μmol, sonicated if not dissolved). The mixture was incubated at 25 °C for 3 h. After quenching the reaction by addition of 0.5 mL of $EtONH_2·HCl$ (0.2 M). The mixture was adjusted at pH 4.0 and incubated at 25 °C for another 3 h. The reaction mixture was analyzed with LCMS to deduce the conversion rate. The major oxime product was collected by HPLC purification for NMR experiments, high-resolution MS characterization or anti-HIV1 assays.

**Expression of protein target RNF4 and SUMO2.** The customized expression plasmids pET 22b/21a (synthesized by Sangon Biotech, Shanghai, Co., Ltd) were transformed into the *E. coli* strain BL21(DE3), which were plated on a selection plate with ampicillin and incubated overnight at 37 °C. A single colony was inoculated into 5 mL LB medium with 100 μg/mL ampicillin. The culture was then incubated at 37 °C overnight with shaking at 220 rpm. Subsequently, 2 mL of the overnight culture was diluted into 200 mL of LB medium with ampicillin. The cells were then incubated at 37 °C with shaking at 220 rpm. When O.D. of the culture reached 0.6, 1 mM IPTG (isopropyl β-D-thiogalactoside) was added and the culture was further incubated at 37 °C with shaking at 220 rpm for another 4 h. Cells were then collected via centrifugation and resuspended in 100 mM phosphate buffer (pH 8.0). The cells were lysed by passing through a homogenizer, and the lysate was centrifuged at 13000 rpm for 30 min. The clarified cellular debris was loaded onto a nickel affinity chromatography column packed with 10 mL Ni-NTA agarose. and the peptide was purified following the manufacturer's instructions.

**Cells and viruses.** TZM-bl cells (Cat No. 8129) were obtained from the NIH AIDS Reagent Program and were cultured in Dulbecco's modified Eagle's medium (DMEM) supplemented with 10% fetal bovine serum at 37°C, 5% $CO_2$ in a humidified incubator. CCR5-tropism HIV-1 lab-adapted strains HIV-1$_{Ba-L}$ was kindly provided by Prof. Una O' Doherty (University of Pennsylvania). Viruses were propagated in Peripheral blood mononuclear cells (PBMCs). The virus titer was determined by a TCID50 assay on TZM-bl cells.

**Cytotoxicity with MTT assay.** 100 μL of TZM-bl suspension culture with cell density at $3 \times 10^5$ viable cells/mL was exposed to different proteins with 6 dilutions in triplicate. Six individual cell cultures without any protein served as negative control, and 6 wells with only medium as blank. The culture was incubated at 37 °C with 5% $CO_2$ for 3 days before the Tetrazolium Dye (MTT) colorimetric assay. Optical density (OD) was measured at 490 nm with reference wavelength at 620 nm using an ELx800 plate reader. $CC_{50}$ (50 % Cytotoxic Concentration) for each protein was calculated and reported.

**Inhibition of HIV-1$_{Ba-L}$ replication in TZM-bl cells.** Each protein was prepared with 5 dilutions, 100 μL of each dilution was transferred to a 96-well plate. 50 μL of TZM-bl culture with cell density at $6 \times 10^5$ viable cells/mL was added, followed by the addition of HIV-1$_{Ba-L}$ dilution with multiplicity of infection (MOI) at 0.03. Cells with only virus served as positive control, and cells without virus served as negative control, and MVC was used as a positive drug control. All experiments were performed in triplicate. After incubation at 37 °C with 5% $CO_2$ for 48 hours, the supernatant was removed, and cells were washed with PBS twice. Then, the cells were lysed by adding 100 μL of luciferase lysis buffer and incubating at 4 °C for 30 min. After the addition of the luminescent substrate with 1/1 ratio, fluorescent signal was recorded using a FLEX Station3 plate reader. $EC_{50}$ was calculated and reported.

**Other detailed synthetic procedures.** All other detailed synthetic procedures, including peptides and CCL4 protein syntheses, synthesis of various alkoxyamines were provided in Supplementary Information.

**Data collection and analysis.** High-performance liquid chromatography-mass spectrometry (HPLC-MS) experiments were performed on a Shimadzu HPLC-MS 2020 system using LC-2020 separation module. The data were collected with an integrated LabSolution v5.8 software and analyzed with MestReNova v12 software. Analytical HPLC traces were recorded with an Agilent 1260 separation module equipped with a DAD detector. The data was collected and analyzed with OpenLAB CDS C 01.07 software. The NMR spectra were recorded on a Bruker Advance 500 spectrometer at 300 K. The data were collected and analyzed with Bruker Topspin v4.1 software. High-resolution ESI mass spectra were obtained on a Waters Vion IMS QTOF spectrometer equipped with an Acquity I-class UPLC separation module. The data was collected and analyzed on a UNIFI v1.9.4 software.

**Reporting summary.** Further information on research design is available in the Nature Research Reporting Summary linked to this article.

## Data availability
All compound characterization data is provided in Supplementary Information. Sequence information of horse myoglobin (Uniprot-KB P68082), human ubiquitin (Uniprot-KB P0CG48), RNF4 (Uniport-KB P78317), SUMO2 (Uniprot-KB P61956) and MIP-1β (Uniprot-KB P13236) was available in uniport databank (uniport.org).

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

## Acknowledgements

Financial support for this work was gratefully received from the National Natural Science Foundation of China (21907064, 22077080, 91753102 and 21672146) and the Interdisciplinary Program of Shanghai Jiao Tong University (YG2020YQ14) and the Strategic Priority Research Program of the Chinese Academy of Sciences (Grant No. XDA12020227). The authors thank X. Li (Instrumental Analysis Center, Shanghai Jiao Tong University) for technical assistance with HRMS experiments. We thank Prof. Yimin Li (Hefei University of Technology) for technical support and Prof. Yongqiang Tu for analytical equipment access.

## Author contributions

P.W. and Y-T.Z. supervised the project. P.W. wrote the manuscript. S.W., Q.Z. and X.C. made contributions to the project design and prepared the supporting information. R-H.L. and L-M.Y. conducted the anti-HIV assays for all MIP-1β variants. These four authors contributed equally to this project. Y.L. and X.L. synthesized some of the peptides used in this project.

## Competing interests

The authors declare no competing interests.
