## [Peer Review File · Nature Communications]

REVIEWER COMMENTS

Reviewer #1 (Remarks to the Author):

This manuscript describes a biomimetic oxidation/transamination strategy to chemo-selectively oxidize the N-terminal amine of a polypeptide chain to a ketone or an aldehyde group with quinone (1) as the oxidant. The reaction is applicable to a range of peptides such as Covid-19 spike protein fragment (Arg), tetracosactide (Ser), myoglobin (Gly) and ubiquitin (Met). Although there have been existing protein modification methods including site-specific protein transamination with different oxidants, this method showed advantages in broad substrate scope and reduced reaction time. I recommend publication in Nat. Commun. after considering the following revisions.

1. Six quinone oxidants have been screened for this oxidation reaction in Figure 2. Since there have been a series of promising bioinspired quinone catalysts (see ref 11 cited in this manuscript), I would like to include more quinone screening data. Q1, Q3, Q5 and Q7 indicated in ref 11 should be tested in this reaction especially for the substrates with low conversions and yields.
2. There are a lot of errors and typos need to be corrected.
 - 1) The numbers of entries did not mean the numbers of oxidants. Please clearly number the oxidants in Figure 2.
 - 2) Compound numbers should be bold. Please bold the numbers in Line 87, 90, 110, 159 and 168.
 - 3) Line 157, "Furthermore, a biotinylated tag was synthesized" should be "Furthermore, a biotinylated tag was synthesized".
 - 4) Numerous "N-terminus" should be "N-terminus". "N" should be Italics. The authors need to check similar English errors thoroughly in the manuscript.

Reviewer #2 (Remarks to the Author):

In this manuscript, the authors report an ortho-quinone enabled oxidation at the N-terminus of peptides and proteins. The reaction renders a carbonyl functionality at the termini that is further utilized for late-stage modification through oxime formation. The authors demonstrated the viability of the method with model peptides, myoglobin, and ubiquitin. Further, they prepared a series of N-residue mutated versions of MIP-1 β using established methods. Finally, they performed anti-HIV-1 assays with the MIP-1 β analogs.

The manuscript's organization of manuscript has some interesting parts, both in the main text and supplementary information, but leaves a considerable improvement scope. For example, the introduction highlights the shortcomings of the previous methods. Some of them are factually incorrect and seems to be forced to highlight a few advantages that are not even rendered by the reported method. The authors can consider rephrasing the introductory paragraphs. In another case, Figure 3 talks about four potential products: oxime, oxime with fragmented side-chain, transaminated product, and Pictet-Spengler product. However, the structure of potential products is provided only in the supplementary information. Further, the LC and MS data do not provide conclusive evidence for the structure of products. A shorter version of the substrate could have simplified the structural investigation by 1D and 2D NMR. Also, MS-MS with the listed peptide could have unambiguously validated the site-of-modification to confirm the method's translation. For N-Pro peptide, the authors mention that the secondary amine does not undergo oxidation. However, do they see any transformation? For reference, they can look at the old paper from Mason and Peterson (J. Biol. Chem. 1955, 212, 485-493; <http://www.jbc.org/content/212/1/485.citation>). The method lacks selectivity (or generality) while it offers diverse pathways with different N-terminus residues. However, it could

still be valuable for a few applications if a proper structural characterization is performed for adducts from all the amino acids. The efforts in this perspective listed in section 8 (page 126, SI) are too preliminary.

Moreover, the application of the reported method to create MIP-1 β analogs is a surprising choice. If one would synthesize a peptide and harness the potential of NCL, it becomes straightforward to install a probe and generate a library of purified material in the same workflow. It is not clear why would anyone synthesize the X1G (A1G, in this case) variant first, and then use two steps to compromise the overall efficiency?

In summary, this manuscript has a few merits but falls short of the bar for publication in Nature Communications due to all the listed reasons.

Reviewer #3 (Remarks to the Author):

This manuscript contains essentially two components – first the authors showed that ortho-quinones can oxidize the N-terminal α -amine to generate an aldo or keto handle that could be further reacted with various hydroxylamine probes to generate a site-specific oxime at the N-terminus. Then the authors used their labeling strategy to modify MIP-1 β to assess how different N-terminal modifications may influence its anti-HIV-1 activity. In my opinion, the authors demonstrated that this method is effective for peptide or small protein labeling (ubiquitin), however, I am less convinced that it can be broadly applied to proteins. Since there are numerous methods at this point that can selectively label the N-terminus of proteins (enzyme-catalyzed, NCL, transamination, etc...) I believe it is important for the authors to show that their method is also robust in this process. In order to strengthen their claim that this method can be a useful tool for protein labeling the authors should provide more convincing evidence for this application as discussed below.

Major criticisms:

1) The authors seem to indicate that this method struggles with proteins that harbor internal cysteines. Although Cys is one of the least abundant residues in the human proteome, most proteins still contain at least one including those that serve important biological functions. The authors worked around this issue by leaving the Cys protected in the synthetic peptide. My question is if they have a general strategy to label expressed proteins that contain free cysteines using this method or is this a significant limitation to its broad utility? If so, it would be important for the authors to show an example for such a protein, present a general strategy to effectively accomplish this, and show that the protein remains functionally intact after the chemistry.

2) The authors also reported the formation of over-oxidized lysines when testing their method with myoglobin even after they changed the pH to 6.0 (~15% undesired product) which could be even worse depending on the hydroxylamine probe used in the second step. A question that should be addressed is if this non-specific reactivity is a general issue when labeling proteins or is this specific to myoglobin? The authors should test this method with other proteins to assess the extent of this non-specific reactivity and present the optimal conditions/procedure (pH, time, concentration of reactants, and temperature) that minimizes the undesired product while also maintaining the intact function of the protein.

Minor comments:

1) The figure legends in the manuscript report reaction conditions that are different than what is

reported in the supplementary file. For example, some of the oxime reaction conditions are reported at pH 4.0 in the manuscript, however, in the supplement it is reported as pH 6.5. Please fix these typos.

2) Line 157, delete "c" in "tagcwas"

Our point-to-point responses

Reviewer 1 (Remarks to the Author)

Comments: This manuscript describes a biomimetic oxidation/transamination strategy to chemo-selectively oxidize the *N*-terminal amine of a polypeptide chain to a ketone or an aldehyde group with Ox1 as the oxidant. The reaction is applicable to a range of peptides such as Covid-19 spike protein fragment (Arg), tetracosactide (Ser), myoglobin (Gly) and ubiquitin (Met). Although there have been existing protein modification methods including site-specific protein transamination with different oxidants, this method showed advantages in broad substrate scope and reduced reaction time. I recommend publication in Nat. Commun. after considering the following revisions.

Corrections and response to reviewer 1:

1. Questions:

Six quinone oxidants have been screened for this oxidation reaction in Fig. 2. Since there have been a series of promising bioinspired quinone catalysts (see Ref. 11 cited in this manuscript), I would like to include more quinone screening data. Q1, Q3, Q5 and Q7 indicated in Ref 11 should be tested in this reaction especially for the substrates with low conversions and yields.

Response:

Oxidant	Structure	Oxidant	Structure	Oxidant	Structure
Ox1		Ox5		Ox9	Ox2		Ox6		Ox10	Ox3		Ox7		Ox11	Ox4		Ox8		Ox12	
We would like to thank this reviewer for his support and constructive suggestions. As the reviewer suggested, we screened another six quinones which gave a total twelve different quinones **Ox1-Ox12** (see table below). Eight of them **Ox1-Ox8** were included in the revised manuscript (Fig. 2). **Ox9-Ox12** were included in the SI (Section 3.1, Page 34-35), which failed to produce desired products. In addition, the tetrachloro-*o*-quinone **Ox3** was also found to have transamination activity, but with some other side reactions.

2. Questions:

There are a lot of errors and typos need to be corrected.

- 1) The numbers of entries did not mean the numbers of oxidants. Please clearly number the oxidants in Fig. 2.
- 2) Compound numbers should be bold. Please bold the numbers in Line 87, 90, 110, 159 and 168.
- 3) Line 157, “Furthermore, a biotinylated tagcwas synthesized” should be “Furthermore, a biotinylated tag was synthesized”.
- 4) Numerous “N-terminus” should be “*N*-terminus”. “N” should be Italics. The authors need to check similar English errors thoroughly in the manuscript.

Response:

We apologize for these mistakes. As suggested by the reviewer, we have revised the manuscripts extensively.

The table of Fig. 2 was revised, oxidants were numbered as **Ox1-Ox12**, which were separated from the other compounds and peptides.

Compound numbers have been revised according to suggestion.

The descriptions of “a biotinylated tagcwas synthesized” have been changed to “a biotinylated tag was synthesized”.

The descriptions of “N-terminus” have been changed to “*N*-terminus” throughout manuscript and SI.

Reviewer 2 (Remarks to the Author)

Comments: In this manuscript, the authors report an ortho-quinone enabled oxidation at the N-terminus of peptides and proteins. The reaction renders a carbonyl functionality at the termini that is further utilized for late-stage modification through oxime formation. The authors demonstrated the viability of the method with model peptides, myoglobin, and ubiquitin. Further, they prepared a series of N-residue mutated versions of MIP-1 β using established methods. Finally, they performed anti-HIV-1 assays with the MIP-1 β analogs.

1. Questions:

The manuscript's organization of manuscript has some interesting parts, both in the main text and supplementary information, but leaves a considerable improvement scope. For example, the introduction highlights the shortcomings of the previous methods. Some of them are factually incorrect and seems to be forced to highlight a few advantages that are not even rendered by the reported method. The authors can consider rephrasing the introductory paragraphs.

Response:

We thank this reviewer for suggestions. We have carefully revised the introduction part.

In paragraph 1, we emphasized that Mason *et al* (Ref. 21) reported an unknown reaction between *o*-quinone and *N*-termini of proteins. This is the starting point of our work. The descriptions of 'Early work was provided by Mason in 1955, who reported that *o*-quinones can react with proteins to form a colored complex *via* the *N*-terminal residue' have been changed to 'Although the early work provided by Mason in 1955 showed that *o*-quinones could react with proteins *via* the *N*-terminal residue, the reaction between *o*-quinone motifs and proteins have not been fully elucidated on a molecular level.'

In paragraph 2, discussion part of previous transamination methods was revised:

The descriptions of 'However, the efficiency of these methods relies on the specific *N*-terminal sequence and prolonged reaction times in the presence of PLP can potentially result in denaturation.' have been changed to 'However, the rate or efficiency of transamination/oxime formation may vary at different *N*-termini or even different amino acid combinations. (Ref. 50, 51) Room for improvements is still remained in the field of *N*-terminal modification strategy. (Ref. 52) Marginally, the salt nature of NaIO₄, PLP or RS salt makes them difficult from aqueous medium to be removed after transamination of peptide or small protein targets, which may potentially interfere the subsequent conjugations (Ref. 53).'

'Transamination and oxime formation strategies and enzymatic labeling methods offer solutions in many instances, but both approaches exhibit varying yields and rates for different amino acid combinations.' (Quotes from Ref. 52) At this point, we believed that our updated discussion was fair.

2. Questions:

In another case, Fig. 3 talks about four potential products: oxime, oxime with fragmented side-chain, transaminated product, and Pictet-Spengler product. However, the structure of potential products is provided only in the supplementary information. Further, the LC and MS data do not provide conclusive evidence for the structure of products. A shorter version of the substrate could have simplified the structural investigation by 1D and 2D NMR. Also, MS-MS with the listed peptide could have unambiguously validated the site-of-modification to confirm the method's translation.

Response:

We thank this reviewer for this constructive suggestion about characterization of peptide products. According to reviewer's suggestion, major products formed from transamination of model peptides with 10 different *N*-termini were characterized by NMR. The results were added in the revised manuscript (Fig. 3) and SI. We assigned each set of ¹H signals with aid of COSY, HSQC and HMBC and included those assignments in the SI. Briefly, we hunted the changing of *N*-terminal H α (disappearing) and H β (shifting) signals between the starting materials and products and hence confirmed the product of Gly **1a** (Page 37-38 in SI), Met **4a** (both geometric isomer, Page 57-60 in SI), Glu **5a** (major isomer, Page 61-63 in SI), Lys **6a** (both geometric isomer, Page 64-67 in SI), Asn **16a** (both geometric isomer, Page 86-89 in SI). In particular, NMR results showed that the two peaks of Lys **6a** were only the geometric isomer. The reaction doesn't touch any of the two side-chain ϵ -amines. The ¹H NMR of major product of Ser **16**, Thr **17**, Trp **18** were recorded and found to be identical to that of **1a** (Page 74-79 in SI). For Asp **7**, structures of both major products were also elucidated from NMR experiments to be **11a** with an oxidative decarboxylation (Page 68-73 in SI). However, although the Δ MS of His **18** adduct was identical to that of Pictet-Spengler product reported (Ref. 39), we found that an oxazine motif was formed between peptide **18** and **Ox1** (Fig. 3B, Page 93-98 in SI). Transaminated products were not confirmed due to the low percentage of abundance and they can be simply pushed towards their corresponding oxime products by extending reaction times. Gln **17** was not optimized due to the natural pyro-Glu post-translational modification. We believe that we now provide solid structural information of the major outcomes of this method.

3. Questions:

For *N*-Pro peptide, the authors mention that the secondary amine does not undergo oxidation. However, do they see any transformation? For reference, they can look at the old paper from Mason and Peterson (*J. Biol. Chem.* 1955, 212, 485-493; <http://www.jbc.org/content/212/1/485.citation>).

Response:

In previous report (Mason and Peterson, *J. Biol. Chem.* **1955**, 212, 485-493), the amine of Pro reacted with quinone *via* a Michael-type reaction mechanism.

In Fig. 3, the quinone oxidant used in transamination is **Ox1**. We designed and screened twelve quinone oxidants with different substitution group (see Fig. 1). The substitution group can fine tune the reactivity of quinone. The *t*Bu and OMe group on 4- and 5- position of **Ox1** blocked the potential attack from nucleophiles. Therefore, such Michael-type reaction is forbidden in this reaction. Our reaction proceeded through a bio-mimic mechanism, starting from formation of an imine between **Ox1** and the α -amine (as Fig. 1, Ref. 11 and Page 166 of SI illustrated). In this context, peptide **19** with *N*-terminal Pro was hardly oxidized as a secondary amine, only trace amount (<2%) of oxime product **19a** was observed (Fig. 3A).

4. Questions:

The method lacks selectivity (or generality) while it offers diverse pathways with different *N*-terminus residues. However, it could still be valuable for a few applications if a proper structural characterization is performed for adducts from all the amino acids.

Response:

We are grateful for reviewer's suggestion. In previous reports, PLP offered a pioneer choice of general transamination (Ref. 38). However, reactivities can vary significantly at different *N*-terminal residues (Ref. 39) or even at different *N*-terminal sequences bearing positive chargers (Witus, L.S. *et al*, *J. Am. Chem. Soc.*, **2010**, *132*, 16812). Moreover, reactivity of PLP varies from batch to batch (Ref 46). Another transamination reagent, RS salt is perfect for targeting *N*-terminal glutamate (Ref. 46). However, the efficiency of this reaction still varies from different peptide sequences (Ref. 51). Our reaction proceeds through a different mechanism. The reactivities of protein *N*-termini via quinone-mediated transamination were quite distinct from PLP and RS salt. It's always essential to expand the arsenal of protein modification methods.

As suggested by the reviewer, full characterizations of most products from model peptides were conducted. Transamination of another two protein examples RNF4 and SUMO was also demonstrated as application of this method.

5. Questions:

The efforts in this perspective listed in section 8 (page 126, SI) are too preliminary.

Response:

Thanks for this suggestion. By LCMS monitoring, a molecular ion of 2-aminophenol **S15** (Page 166-199 in SI) was detected before decomposing. In the enzymatic catalytic cycle of CuAOs, *ortho*-quinones co-factors were extensively characterized (Ref. 3). Many researchers adapted this concept for the development of quinone-catalyzed transamination methodologies (Ref. 4-

11) for small molecules. We adapted **Ox1** from the rest of quinones owing to its reactivity and selectivity. The mechanism of transamination with **Ox1** on small molecule amine was extensively discussed by previous reports (Ref. 11). Combined with the existed evidence, we were confident about our purposed mechanism as illustrated below. The key steps involved the formation of an imine between quinone and amine, followed by extraction of α -proton by another quinone keto group to generate the second imine intermediate that can be hydrolyzed into glyoxylate moiety. Changes were made in Section 8 of SI. Molecular ion of 2-aminophenol **S15** was observed.

6. Questions:

Moreover, the application of the reported method to create MIP-1 β analogs is a surprising choice. If one would synthesize a peptide and harness the potential of NCL, it becomes straightforward to install a probe and generate a library of purified material in the same workflow. It is not clear why would anyone synthesize the X1G (A1G, in this case) variant first, and then use two steps to compromise the overall efficiency?

Response:

We are grateful for reviewer's suggestion. MIP-1 β proteins were synthesized by NCL, followed by late-stage diversification of these proteins to 17 analogs. The late-stage modification of complex peptides is a powerful tool for protein diversification in a step-economic manner. In contrast, if these 17 proteins would be prepared *via de novo* synthesis, 17 separate ligations and folding reactions had to be employed to assemble each of them, as well as the purification steps in between and after. We believed that it is more efficient to diversify these proteins *via* late-stage modification strategy.

With all those amendments in the revised manuscript, we hope the reviewer could kindly reconsider our work.

Reviewer 3 (Remarks to the Author)

Comments: This manuscript contains essentially two components – first the authors showed that *ortho*-quinones can oxidize the *N*-terminal α -amine to generate an aldo or keto handle that could be further reacted with various hydroxylamine probes to generate a site-specific oxime at the *N*-terminus. Then the authors used their labeling strategy to modify MIP-1 β to assess how different *N*-terminal modifications may influence its anti-HIV-1 activity. In my opinion, the authors demonstrated that this method is effective for peptide or small protein labeling (ubiquitin), however, I am less convinced that it can be broadly applied to proteins. Since there are numerous methods at this point that can selectively label the *N*-terminus of proteins (enzyme-catalyzed, NCL, transamination, *etc*) I believe it is important for the authors to show that their method is also robust in this process. In order to strengthen their claim that this method can be a useful tool for protein labeling the authors should provide more convincing evidence for this application as discussed below.

1. Questions:

The authors seem to indicate that this method struggles with proteins that harbor internal cysteines. Although Cys is one of the least abundant residues in the human proteome, most proteins still contain at least one including those that serve important biological functions. The authors worked around this issue by leaving the Cys protected in the synthetic peptide. My question is if they have a general strategy to label expressed proteins that contain free cysteines using this method or is this a significant limitation to its broad utility? If so, it would be important for the authors to show an example for such a protein, present a general strategy to effectively accomplish this, and show that the protein remains functionally intact after the chemistry.

Response:

We thank the reviewer for pointing out this issue. With regards to proteins containing Cys residues, we introduced photo-labile protecting strategy. The transamination of expressed RNF4 32-133 fragment (three free Cys residues) and SUMO2 protein (one Cys residue) was incorporated (Fig. 5). Cys residues of RNF4 and SUMO2 were temporarily protected with photo-labile 2-nitropiperonyl group, which can be removed readily by irradiation with UV light ($\lambda=365$ nm) within 30 min after the transamination reaction. All three steps (protection,

transamination and deprotection) were conducted in pH 6.5 buffer with simple purifications such as extraction or dialysis, without using strong reductant such as DTT or TCEP for deprotection. In addition, a comparison of CD spectra of SUMO2 protein before and after the set of reactions was presented in Page 121 of SI to show that the protein remained its tertiary structure after modifications. Furthermore, our reaction remained functionally intact with existed disulfide bridges which is clearly presented in the CCL4 protein library section. Related information was added in manuscript as well as Section 3.8 and 3.9 in SI.

2. Questions:

The authors also reported the formation of over-oxidized lysines when testing their method with myoglobin even after they changed the pH to 6.0 (~15% undesired product) which could be even worse depending on the hydroxylamine probe used in the second step. A question that should be addressed is if this non-specific reactivity is a general issue when labeling proteins or is this specific to myoglobin? The authors should test this method with other proteins to assess the extent of this non-specific reactivity and present the optimal conditions/procedure (pH, time, concentration of reactants, and temperature) that minimizes the undesired product while also maintaining the intact function of the protein.

Response:

We are grateful for the reviewer's suggestions. During the model study, pH 6.5 was selected because this condition gave the highest reactivity and only trace amount of Lys ϵ -amine oxidation was observed. This condition worked perfectly as demonstrated by other peptides and proteins included in this paper. Myoglobin contained 20 Lys residues in its sequence. Thus, slight lowering of pH was chosen to minimize the occurring of this undesired side-reaction. Importantly, oxime formation is a separated reaction from transamination. In the case of myoglobin, syntheses of ethylated myoglobin **23a** and biotinylated variant **23b** were conducted with same pot of transamination but adding different alkoxyamine, which meant that transamination was indeed achieved by 70% target conversion (~15% over oxidation). Unfortunately, formation of oxime with biotinylated alkoxyamine **S5** resulted in decreasing of the total conversion rate, which was not caused from further oxidation of Lys, as the doubly biotinylated variants (with a molecular ion m/z 17505) was hardly observed (in SI Page 108,

Fig. 217-218). Nevertheless, this result also demonstrated one advantage of this method, that is the easy removal of excess Ox1 or other resulting small molecule byproducts by organic extraction to prevent possible interference in the subsequent conjugations.

Due to the non-specific micro-environment of each protein, reactivities of the *N*-termini of proteins differ from case to case. Reaction optimization is necessary because none of the current methods could offer a general solution for all *N*-terminal transamination. We believe that our method can still serve as a powerful complementary method for general *N*-terminal transamination by offering an alternative method.

Two new proteins (RNF4 and SUMO) were tested, giving the transaminated products with good yields. As the oxidant and resulting small molecule can be easily removed with extraction, no other purification was necessary between transamination and photo-irradiation.

With all those amendments, we hope our extended work can satisfy the critiques of this reviewer.

Minor comments:

3. Questions:

The figure legends in the manuscript report reaction conditions that are different than what is reported in the supplementary file. For example, some of the oxime reaction conditions are reported at pH 4.0 in the manuscript, however, in the supplement it is reported as pH 6.5. Please fix these typos.

Response:

Corrections were made as the reviewer suggested.

Oxime formations of peptidyl products with EtONH₂ were conducted at pH 4 to accelerate the reaction. In protein examples, pH 6.5 was chosen when the inert alkoxyamine was used during oxime formation because this pH has less chance to damage the proteins.

4. Questions:

Line 157, delete “c” in “tagcwas”

Response:

We apologize for this mistake. Corrections were made as the reviewer suggested.

REVIEWERS' COMMENTS

Reviewer #1 (Remarks to the Author):

The revision of the paper titled "Modification of N-Terminal α -Amine of Proteins via Biomimetic ortho-quinone-mediated Oxidation" pertinently addresses the reviewers' previous comments. Screening data of more quinones, major products formed from transamination of model peptides with different N-termini, and the results of transamination with proteins bearing free-Cys residues were all supplemented in the revised manuscript and SI. The revisions made the description more accurate and conclusion stronger thus improve the scientific value of the study. Given those improvements, I would suggest the publication of this manuscript in Nat. Commun.

Reviewer #2 (Remarks to the Author):

The authors have done an excellent job in addressing most of the concerns satisfactorily. For a few unaddressed points, they have provided a logical scientific justification. Hence, I believe that the manuscript can be considered for publication in Nature Communications.

Reviewer #3 (Remarks to the Author):

This manuscript presents a strategy to site-selectively label the N-terminus of proteins and peptides using an ortho-quinone based strategy to oxidize the α -amine of the N-terminus which is then further reacted with hydroxylamine probes to generate an N-terminal oxime. Then the authors used their labeling strategy to modify MIP-1 β to assess how different N-terminal modifications may influence its anti-HIV-1 activity.

The authors have satisfied my concerns with the addition of two proteins (SUMO2 and RNF4) and a photo-cleavable cysteine protecting strategy that makes this strategy more universally applicable to proteins/peptides that contain free cysteine(s). This manuscript is much improved and I recommend for it to be published.